# Geospatial Analyses of Recent Household Surveys to Assess Changes in the Distribution of Zero-Dose Children and Their Associated Factors before and during the COVID-19 Pandemic in Nigeria

**DOI:** 10.3390/vaccines11121830

**Published:** 2023-12-08

**Authors:** Justice Moses K. Aheto, Iyanuloluwa Deborah Olowe, Ho Man Theophilus Chan, Adachi Ekeh, Boubacar Dieng, Biyi Fafunmi, Hamidreza Setayesh, Brian Atuhaire, Jessica Crawford, Andrew J. Tatem, Chigozie Edson Utazi

**Affiliations:** 1WorldPop, School of Geography and Environmental Science, University of Southampton, Southampton SO17 1BJ, UK; i.d.olowe@soton.ac.uk (I.D.O.); hmtc1u18@soton.ac.uk (H.M.T.C.); a.j.tatem@soton.ac.uk (A.J.T.); c.e.utazi@soton.ac.uk (C.E.U.); 2Department of Biostatistics, School of Public Health, College of Health Sciences, University of Ghana, Accra P.O. Box LG13, Ghana; 3School of Mathematical Sciences, University of Southampton, Southampton SO17 1BJ, UK; 4Sydani Group, Abuja, Nigeria; adachi.ekeh@sydani.org; 5Gavi, The Vaccine Alliance, Abuja, Nigeria; bdieng@gmail.com; 6National Bureau of Statistics, Abuja, Nigeria; biyifafunmi@nigerianstat.gov.ng; 7Gavi, The Vaccine Alliance, Geneva, Switzerland; hsetayesh@gmail.com (H.S.); batuhaire@gavi.org (B.A.); jcrawford@gavi.org (J.C.); 8Department of Statistics, Nnamdi Azikiwe University, Awka PMB 5025, Nigeria

**Keywords:** MCV1 coverage, DTP1 coverage, composite coverage, zero-dose prevalence, Demographic and Health Surveys, Multiple Indicator Cluster Survey, Bayesian geostatistical modelling, Bayesian multilevel modelling

## Abstract

The persistence of geographic inequities in vaccination coverage often evidences the presence of zero-dose and missed communities and their vulnerabilities to vaccine-preventable diseases. These inequities were exacerbated in many places during the coronavirus disease 2019 (COVID-19) pandemic, due to severe disruptions to vaccination services. Understanding changes in zero-dose prevalence and its associated risk factors in the context of the COVID-19 pandemic is, therefore, critical to designing effective strategies to reach vulnerable populations. Using data from nationally representative household surveys conducted before the COVID-19 pandemic, in 2018, and during the pandemic, in 2021, in Nigeria, we fitted Bayesian geostatistical models to map the distribution of three vaccination coverage indicators: receipt of the first dose of diphtheria-tetanus-pertussis-containing vaccine (DTP1), the first dose of measles-containing vaccine (MCV1), and any of the four basic vaccines (bacilli Calmette-Guerin (BCG), oral polio vaccine (OPV0), DTP1, and MCV1), and the corresponding zero-dose estimates independently at a 1 × 1 km resolution and the district level during both time periods. We also explored changes in the factors associated with non-vaccination at the national and regional levels using multilevel logistic regression models. Our results revealed no increases in zero-dose prevalence due to the pandemic at the national level, although considerable increases were observed in a few districts. We found substantial subnational heterogeneities in vaccination coverage and zero-dose prevalence both before and during the pandemic, showing broadly similar patterns in both time periods. Areas with relatively higher zero-dose prevalence occurred mostly in the north and a few places in the south in both time periods. We also found consistent areas of low coverage and high zero-dose prevalence using all three zero-dose indicators, revealing the areas in greatest need. At the national level, risk factors related to socioeconomic/demographic status (e.g., maternal education), maternal access to and utilization of health services, and remoteness were strongly associated with the odds of being zero dose in both time periods, while those related to communication were mostly relevant before the pandemic. These associations were also supported at the regional level, but we additionally identified risk factors specific to zero-dose children in each region; for example, communication and cross-border migration in the northwest. Our findings can help guide tailored strategies to reduce zero-dose prevalence and boost coverage levels in Nigeria.

## 1. Introduction

Vaccination is often regarded as one of the most successful and cost-effective public health interventions, saving millions of lives each year and guaranteeing global well-being and development [1]. Despite this, many children, especially those living in low- and middle-income countries (LMICs), continue to miss out on life-saving vaccines even though there have been increased efforts globally to improve vaccination coverage and reduce zero-dose prevalence [2]. Before the pandemic in 2019, 18.4 million children globally did not receive all three recommended doses of the DTP vaccine, and of those, 70% (12.9 million) were zero-dose children, defined as those who did not receive any dose of the DTP vaccine [2]. In 2020, these figures increased to 22 million children and 73% (16 million), respectively, due to the disruptions to immunization services caused by the coronavirus disease 2019 (COVID-19) pandemic [2,3,4]. These disruptions continued in 2021, resulting in 24 million being under-vaccinated and about 18 million being zero dose, with about 62% [5,6] of the estimated zero-dose children found to be living in 10 LMICs, including Nigeria. However, in 2022, a partial recovery in global DTP vaccination coverage was recorded, with the number of zero-dose children decreasing to 14.3 million, evidencing concerted efforts within countries to reach zero-dose children [2]. 

Zero-dose children often live in marginalized or underserved communities characterised by poverty, a lack of access to basic health services, overcrowding, poor sanitation practices, and conflict [7,8,9,10]. These characteristics, combined with other health-related, socioeconomic, demographic, and gender-related factors, cause substantial disparities in the distribution of zero-dose children within countries [8]. Reaching these at-risk populations, therefore, requires a timely and accurate evidence base regarding their sizes, geographic distribution, and other characteristics, to support country-tailored strategies and interventions. Also, with recovery from the pandemic being uneven and much slower in LMICs [11], understanding any changes in vulnerabilities due to disruptions to both routine immunization and vaccination campaigns can help with planning effective mitigation strategies and strengthening immunization services to reach zero-dose children. Administrative data are regularly collected in many LMICs through platforms such as the District Health Information System version 2 (DHIS2) [12,13]. However, due to limitations such as numerator and denominator errors (e.g., incomplete reporting, inaccurate the aggregation of numerators, and mismatches between numerator and denominator estimates due to migration and the bypassing of health facilities), these often have coverage values that cannot reliably inform spatially detailed heterogeneities in the coverage and identification of zero-dose children. Household surveys, on the other hand, tend to produce more reliable estimates of coverage, but these are usually designed to be representative at coarse spatial scales, necessitating the use of geospatial modelling approaches to produce coverage estimates at fine spatial scales and for operationally relevant areas, e.g., districts, which are then integrated with population data to assess zero-dose prevalence [14,15,16]. Moreover, survey questionnaires include several modules that assess different characteristics of the participants, making the data ideal for evaluating correlates of non-vaccination to further inform targeted interventions. Undoubtedly, addressing zero-dose prevalence is critical to achieving the WHO’s Immunisation Agenda 2030 target of a 50% reduction in zero-dose children by 2030 and promises to “leave no one behind”, as well as targets within the Sustainable Development Goals [7,17] and Gavi, the Vaccine Alliance’s 2021–2025 Strategy [7,18]. 

Nigeria has one of the largest cohorts of un- and under-vaccinated children globally, with 2.3 million and 3 million children estimated to not have received any dose of the DTP and MCV vaccines, respectively, in 2022 [2]. Before the pandemic in 2019, the routine coverage of essential vaccines such as DTP1 and MCV1 was estimated to be 72% and 58%, respectively. In 2022, although global coverage levels showed some recovery following the pandemic, routine coverage remained suboptimal in Nigeria, standing at 70% and 60%, respectively, for both basic vaccines [2]. As a result, Nigeria has continued to experience measles outbreaks, with a resurgence of diphtheria outbreaks in 2023 [19]. Utazi et al. [20] found that despite repeated measles vaccination campaigns, measles’ incidence was related to routine immunization (RI) coverage. Over the years, there has been a persistent north–south divide in the vaccination coverage in Nigeria, with the northern regions having poorer coverage levels and often higher rates of disease incidence [20,21]. Many studies have also identified several demand- and supply side factors such as low rates of maternal education, belonging to certain religious groups, poor maternal access to and utilization of health services, the poor attitude of health workers, staff shortages, poor conditions at health facilities and vaccine stockouts [22,23,24,25], and geographic factors such as remoteness and living in an urban slum [7,26], as being responsible for poor vaccine uptake and heterogeneities in the distribution of vaccination coverage within the country. The first case of the SARS-CoV-2 virus was recorded on 27 February 2020 [27] in Lagos State, Nigeria, following which the government launched a response to the pandemic, including a lockdown from 30 March to 15 May 2020 [28]. COVID-19 vaccination began on 5 March 2021, which saw significant shifts of priorities and resources from vaccination services to the COVID-19 response [29]. These and other interventions are also thought to have further impacted immunization services negatively in the form of reduced access to vaccination, decreases in vaccine demand and uptake, the cessation of outreach services, and the postponement of vaccination campaigns [3,20,30,31,32]. These challenges call for innovative approaches and intensified efforts to identify and reach zero-dose children in Nigeria. 

Against this backdrop, our study aimed to estimate changes in the spatial distribution of zero-dose children and the associated risk factors before and during the COVID-19 pandemic in Nigeria, with a view to assessing the impact of the pandemic on immunization service delivery in the country, which can help consolidate mitigation and other strategies required to boost coverage beyond pre-pandemic levels. We analyzed three outcomes/indicators using data from two household surveys implemented before and during the COVID-19 pandemic in Nigeria. We defined a zero-dose child for each outcome as a child aged 12–23 months who had not received DTP1 (i.e., DTP or PENTA zero dose) or MCV1 (MCV zero dose) or any dose of the four basic vaccines—BCG, OPV0, DTP1, and MCV1—(composite zero dose). Due to data constraints, our study considered only demand-side factors or reasons for non-vaccination. 

## 2. Materials and Methods

### 2.1. Data and Sources 

We utilized data from two recent household surveys conducted in Nigeria, namely the 2018 Demographic and Health Survey (DHS) [33] and the 2021 Multiple Indicator Cluster Survey—National Immunization Coverage Survey (MICS-NICS) [34]. We also assembled geospatial covariate data obtained from various sources and relevant geospatial population data. To ensure respondent confidentiality, the cluster-level geographical coordinates were displaced up to 2 km in urban areas and up to 5 km in rural areas. We present detailed descriptions of these data sources in this section.

#### 2.1.1. 2018 Nigeria Demographic and Health Survey (DHS)

The 2018 Nigeria DHS was conducted between August and December 2018. The survey was designed to be representative at the national, zonal, and state levels (including the Federal Capital Territory) and for urban and rural areas. It employed a two-stage stratified sampling design with stratification achieved by separating each of the 36 states and the Federal Capital Territory (FCT) into urban and rural areas. The first and second stages of the sampling design involved the selection of enumeration areas (EAs) or survey clusters with a probability proportional to their size from each stratum, using a national sampling framework and the selection of households at random from household lists within the selected clusters. Detailed information on the methods employed in the survey is published elsewhere [33].

The survey was implemented in a total of 1389 clusters, with 11 of the 1400 clusters selected initially dropped due to security reasons. Also, in Borno State, only 11 of the 27 local government areas were considered in the survey due to high insecurity. Data on children between age 12–23 months were extracted for this study. Information on routine vaccination coverage obtained from both home-based records (or vaccination cards) and through maternal/caregiver recall were included in our study, as in previous studies [7,23]. 

#### 2.1.2. 2021 Nigeria Multiple Indicator Cluster Survey—National Immunization Coverage Survey (MICS-NICS) 

UNICEF implements MICS surveys to collect globally comparable data on several indicators relating to the situation of women and children within countries. On the other hand, the NICS surveys are implemented by the Nigerian government to provide reliable estimates of the indicators of vaccination coverage used to evaluate the performance of the vaccination program. The MICS survey was integrated with NICS for the first time in Nigeria during its 5th round in 2016–2017, paving the way for the joint implementation of both surveys in the current round in 2021. 

Field work for the 2021 MICS-NICS took place between September and December 2021. Similar to the 2018 NDHS, the survey had a two-stage stratified sampling design and was also representative at the national, zonal, and state levels and for urban and rural areas. Details of the sampling methodology are provided in the survey report [34]. The MICS had a target sample size of 1850 clusters. A supplemental sample of 337 clusters was selected for the NICS to increase the combined sample of children and the precision of the vaccination coverage indicators, resulting in a total of 2187 clusters for the MICS-NICS. About 128 of the combined sampled clusters were inaccessible and could not be visited during the survey. Also, in Borno State, sampling took place in only 7 (out of 27) accessible local government areas, in which 29% of the total population of the state resided. As with the 2018 DHS, we extracted all relevant information on routine vaccination coverage for our study for children aged 12–23 months. 

As we show in Appendix A, children aged 12–23 months in the MICS-NICS survey were born between September 2019 and September 2020. Among these, those born after January 2020 became eligible to receive BCG, OPV0, and DTP1 vaccinations during the pandemic, whereas the entire birth cohort became eligible to receive MCV1 within the pandemic period. Also, the first and second waves of the pandemic, which peaked in July 2020 and February 2021, respectively, overlapped considerably with the time intervals during which the MICS-NICS birth cohort became eligible for all four vaccines included in our study. This demonstrates that the analysis carried out using the 2021 MICS-NICS is ideal for assessing the impact of the COVID-19 pandemic on immunization service delivery within the country. Also, since data collection for the 2018 DHS took place before the pandemic, that survey is suitable for assessing the performance of the vaccination program before the pandemic. However, because both surveys were implemented independently and not as rolling/repeated surveys, there could be sampling and other methodological differences that could impact the comparisons between both surveys, which is a potential limitation of our study. 

#### 2.1.3. Outcome Indicators of Zero-Dose Children Included in the Study 

To assess the changes in zero-dose prevalence and the associated risk factors before and during the pandemic in Nigeria, our study considered binary indicators of the receipt of DTP1 (PENTA1) (yes = 1, no = 0), receipt of MCV1 (yes = 1, no = 0), and receipt of any of the four basic vaccines—BCG, OPV0, DTP1, and MCV1—as a composite coverage indicator (yes = 1, no = 0) among children aged 12–23 months. We note that both BCG and OPV0 are birth doses, while DTP1 and MCV1 are administered at the age of 6 weeks and from the age of 9 months, respectively, according to Nigeria’s immunization schedule [35]. For all the 3 indicators, we extracted data on 5459 children and 6393 children aged 12–23 months from the 2021 MICS and the 2018 DHS, respectively, for our analysis. 

At the cluster level, we aggregated the individual level data to produce the numbers of children surveyed, numbers vaccinated, and empirical proportions of children vaccinated. In each case, we obtained the (displaced) geographical (i.e., longitude and latitude) coordinates of the survey clusters. These cluster-level data are displayed in Figure 1 for both surveys.

#### 2.1.4. Independent Variables and Geospatial Covariate Data 

Following previous studies [7,23], we included the individual child, mother, household, and community characteristics as risk factors for being zero-dose [7,23]. These are the sex of the child, skilled birth attendance, mother’s receipt of tetanus toxoid vaccination, mother’s antenatal care visits, maternal age and mother’s marital status, maternal education, religion, access to media and phone/internet, land ownership, health insurance, ethnicity, sex of household head, and household wealth. Other covariates considered are maternal access to a bank account, household size and length of stay, and place (urban/rural) and region of residence. These variables were considered due to their data availability in both the 2018 DHS and 2021 MICS-NICS surveys. Detailed definitions of the variables are provided in Appendix A. 

The geospatial covariates considered include travel time to the nearest health facility, distance to conflict areas, poverty index, number of wet days, daytime land surface temperature, livestock density index, slope, enhanced vegetation index (EVI), distance to coastline, distance to the edge of cultivated areas, proximity to national borders, and proximity to protected areas. Consideration of these covariates was informed by their use in previous studies [15,20,36] to model and predict various indicators of vaccination coverage. These covariates were processed as detailed in previous work [15,20,36,37] to produce 1 × 1 km raster layers and cluster-level data using the geographical coordinates from each of the surveys. Some of these covariate layers are displayed in Appendix A and detailed descriptions are provided in Appendix A. The classifications of the cluster-level values of some of the covariates, for use in the multilevel analyses, are also shown in Appendix A. 

Furthermore, for our multilevel analyses (see model (1)) using each survey, we calculated the tertiles of the distribution of the extracted cluster-level data and used these to group the (continuous) values of the covariates into three classes, namely, lower, medium, and higher, which were used in the analyses (see Appendix A) together with the survey-derived covariates discussed previously. However, for the geospatial models, the included covariates were on their original continuous scale (except where these were log-transformed prior to model-fitting). As in previous studies [16,36,37], for each survey and indicator, we implemented a detailed covariate selection process to determine the best combination of the geospatial covariates to be included in the geospatial analyses, using model (2). The covariate selection process involved various steps to check the relationships between the geospatial covariates and the coverage indicators, resolve the problem of multicollinearity and then choose the best set of covariates using stepwise regression (backward elimination based on Akaike Information Criterion (AIC)) in a non-spatial framework. The final step of the covariate selection process involved creating a uniform set of covariates for modelling all the indicators for both time periods to enhance comparability. 

#### 2.1.5. Population Data

We obtained 1 × 1 km estimates of the numbers of children aged under 1 year old in 2017 and 2020 (corresponding to the birth cohorts included in our analyses) from WorldPop [38], adjusted to the United Nations Population Division (UNPD) estimates at the national level for both time periods [39]. These data were used in our work to calculate zero-dose estimates through integration with the coverage maps, and as weighting layers when aggregating grid-level coverage estimates to the administrative level. 

### 2.2. Statistical Analysis 

#### 2.2.1. Descriptive and Bivariate Analysis

Descriptive analyses were performed using individual-level data for each survey to estimate the frequencies and corresponding proportions for each indicator at the national level, as a precursor to the multilevel analyses. 

#### 2.2.2. Multilevel Model

We fitted Bayesian multilevel random intercept logistic regression models, accounting for individual-, household-, community-, and stratum-level variation, to estimate the relationships between each outcome variable (i.e., odds of DTP1, MCV1 and composite coverage) and the covariates/risk factors for zero dose. 

Let *i* indicate a child aged 12–23 months residing in household *j*, community/cluster *k* and stratum *l* (there were 37 strata in MICS-NICS and 74 in DHS). Also, let $x_{ijkl}$ be a vector of the associated covariates. The multilevel model is given by
$$Y_{ijkl}|p_{ijkl}∼Binomial\left(1,p_{ijkl}\right),\;i=1,\ldots ,n_{jkl},\;j=1,\ldots ,n_{kl},\;k=1,\ldots ,\;n_{l},\;l=1,\ldots ,\;L,$$
(1)$$log\left(\frac{p_{ijkl}}{1-p_{ijkl}}\right)=\gamma_{0}+x_{ijkl}{}^{\prime }\beta +\alpha_{jkl}+\upsilon_{kl}+\tau_{l},\alpha_{jkl}\sim N(0,\;\sigma_{\mu }^{2}),\;\upsilon_{kl}\sim N(0,\;\sigma_{v}^{2}),\;\tau_{l}\sim N(0,\;\sigma_{\tau }^{2})$$
where $Y_{ijkl}$ denotes a binary response (or vaccination status; coded as 1—vaccinated and 0—unvaccinated) for child ijkl, $p_{ijkl}$ represents the corresponding odds of DTP1, MCV1 or zero-dose vaccination, $\gamma_{0}$ is the overall intercept, β is a vector of the associated regression coefficients for the covariates $x_{ijkl}$, $\tau_{l}$ is the residual random effect for stratum *l*, $\upsilon_{kl}$ is the residual random effect for community (or survey cluster) *k* located in stratum *l*, and $\alpha_{jkl}$ is the random effect of household *j* within community *k* located in stratum *l*. The quantities $\alpha_{jkl}$, $\upsilon_{kl}$, and $\tau_{l}$ are assumed to be identically and independently normally distributed with the zero means and variances $\sigma_{\alpha }^{2}$, $\sigma_{v}^{2}$, and $\sigma_{\tau }^{2},\;$respectively [40]. We note that the individual level (level 1) residual is assumed to follow standard logistic distribution with variance expressed as$\;\frac{\pi^{2}}{3}≅$ 3.29 [41].

We applied model (1) to both surveys to identify the significant associated risk factors for zero dose, using all three indicators at the national level. In addition, we applied the model to examine regional variation in the risk factors associated with zero dose by subsetting the data to the north central, north east, north west and southern regions of the country. The three geopolitical zones in the southern part of the country (i.e., the southeast, south-south, and southwest regions) were combined in the regional analysis due to insufficient sample sizes. Also, a reduced set of risk factors were considered in the regional analyses to increase the samples sizes within the categories of the risk factors in each region. 

The Bayesian models were fitted using the integrated nested Laplace approximation (INLA) approach, implemented in the R-INLA package. The default priors in R-INLA were assigned to both the fixed- and random-effect parameters in the models [42]. Following model-fitting, we calculated the adjusted odds ratios and their associated 95% credible intervals to evaluate the significance of the associations between the risk factors and the odds of zero dose. 

#### 2.2.3. Geostatistical Model 

To predict each of the outcome indicators on a 1 × 1 km grid, we applied a Bayesian geostatistical model to the aggregated cluster-level data from each survey. Let $Y(s_{i})$ denote the number of children who received DTP1, MCV1, or any of the four basic vaccines (BCG, OPV0, DTP1, and MCV1) out of a total of $m(s_{i})$ children drawn from each sampled cluster location $s_{i},\;i=1,\;2,\;\ldots ,\;n$, and $p\left(s_{i}\right)$ represent the corresponding unknown true vaccination coverage. Also, let $x(s_{i})$ denote a vector of the geospatial covariate information for location $s_{i}$. The geostatistical model assumes that $Y(s_{i})$ follows the binomial probability distribution given by
$$Y(s_{i})|p\left(s_{i}\right)\;\sim \;Binomial\left(m\left(s_{i}\right),\;p\left(s_{i}\right)\right),$$
(2)$$log\;\left(\frac{p\left(s_{i}\right)}{1-p\left(s_{i}\right)}\right)=\gamma_{0}+x{\left(s_{i}\right)}^{\prime }\beta +\omega \left(s_{i}\right)+\epsilon \left(s_{i}\right)$$
where $\gamma_{0}$ is an intercept parameter, β is a vector of the regression coefficients corresponding to $x(s_{i})$, and $\omega (s_{i})$ is a spatially structured random effect and follows a zero-mean Gaussian process with variance $\sigma^{2}$ and a covariance function, $\Sigma_{\omega }$. There are various parametric families for $\Sigma_{\omega }$ [43]. In the current analysis, we assumed the Matérn class of covariance functions [44] given by
$$\Sigma_{\omega }\left(s_{i},s_{j}\right)=\frac{\sigma^{2}}{2^{\upsilon -1}\Gamma (\upsilon )}{\left(\kappa ||s_{i}-s_{j}||\right)}^{\upsilon }K_{\upsilon }(\kappa ||s_{i}-s_{j}||)$$
where the notation $\left\|.\right\|$ denotes the Euclidean distance between locations $s_{i}$ and $s_{j}$, $\sigma^{2}$ is the variance of the spatial field $\omega (s_{i})$ as noted earlier, υ is a smoothness parameter, κ>0 is a scaling parameter related to the range $r=\frac{\sqrt{8\upsilon }}{\kappa }$—the distance at which spatial correlation is negligible or approaches 0.1 and $K_{\upsilon }(.)$ is the modified Bessel function of the second kind and order υ>0. The smoothness parameter υ was set to 1 for the purpose of identifiability, as recommended [45]. Lastly, $\epsilon (s_{i})$ is an iid Gaussian random effect with mean 0 and variance, $\sigma_{\epsilon }^{2}$, capturing non-spatial residual variation. 

The geostatistical model was fitted using the Integrated Nested Laplace Approximation—Stochastic Partial Differential Equations (INLA-SPDE) approach, implemented in the R-INLA package [45,46]. The predictive performance of all the fitted models were assessed using approaches discussed in previous work [16].

To aggregate the 1 × 1 km grid-level estimates to the district and other administrative levels, we computed the areal estimates as population-weighted averages of the corresponding indicators (i.e., DTP1, MCV1, or composite coverage) taken over all the grid cells falling within the administrative area, as in previous work [16]. We note that this is a common approach to handling point-to-area misalignment when mapping health and development indicators [16,47]. 

## 3. Results

### 3.1. Outcome Indicators of Vaccination Coverage 

The national-level coverage estimates for DTP1, MCV1, and the composite coverage indicator were 64.8% (95% CI: 63.6–66.0%), 53.4% (95% CI: 52.1–54.6%), and 73.8% (95% CI: 72.7–74.8%), respectively, for the 2018 DHS. For the 2021 MICS-NICS, the national-level coverage estimates were 71.0% (95% CI: 69.8–72.2%), 61.1% (95% CI: 59.8–62.4%), and 79.4% (95% CI: 78.3–80.4%), respectively, for DTP1, MCV1, and the composite coverage indicator. Generally, the coverage estimates appeared to be higher for the 2021 MICS-NICS than the 2018 DHS for all the three indicators (Appendix A). 

### 3.2. 1 km × 1 km Modelled Estimates of Coverage and Associated Uncertainties before and during the Pandemic

Predicted coverage estimates and associated uncertainties for children aged 12–23 months in 2018 and 2021 are presented in Figure 2 and Appendix A for DTP1, MCV1, and the composite coverage indicator. These maps show broadly similar patterns in coverage in both years, although coverage seemed relatively higher in some areas (e.g., parts of the northwest) in 2021. There are substantial geographical differences in coverage when examining DTP1 and MCV1 coverage, with a clear north–south divide for both vaccines and years. As expected, the coverage of the composite indicator is generally higher than DTP1 and MCV1 and areas of low coverage are also concentrated more in the northern areas for this indicator, as well as some southern coastal areas and some areas in Cross River state. Importantly, there are substantial overlaps in low coverage areas across all three indicators in both time periods, suggesting a persistent lack of access to vaccination services in these areas. These low coverage areas are more pronounced in the Sokoto, Zamfara, Yobe, and Kwara states, and parts of the Bauchi, Gombe, and Taraba states. In both the Sokoto and Zamfara states, the poorest coverage levels were observed in areas such as Tangaza, Sangiwa, Naman Goma, Tureta, Anka, Ramfashi, Maru, Bungudu, Yar-Mahanga, and Maholo, among others, when examining the interactive web-based maps (https://data.worldpop.org/repo/lf/visual/justice/all_indicators_coverage_final/ (accessed on 28 September 2023)). 

The corresponding uncertainty maps showed standard deviations less than 0.33 for the predicted vaccination coverage estimates for 2018 DHS and less than 0.36 for the 2021 MICS-NICS, suggesting low uncertainties around the predicted coverage estimates in both years (Appendix A). 

### 3.3. District-Level Estimates of the Numbers of Zero-Dose Children before and during the Pandemic

Figure 3 presents district-level estimates of the numbers of zero-dose children for the three coverage/zero-dose indicators before and during the pandemic, in 2018 and 2021, respectively. In general, the spatial distributions of the zero-dose estimates are identical across all three indicators in both time periods. The district-level zero-dose estimates exhibit a clear north–south divide similar to the coverage estimates, with children residing in the northern districts being at higher risk of zero dose for all three indicators compared to their counterparts in the south. However, there are also clusters of districts with relatively higher numbers of zero-dose children in the south (in the Lagos and Ogun states). As expected, there is a substantial overlap between the low coverage areas and areas with higher numbers of zero-dose children, although there are a few exceptions. For example, some districts in the southern coastal areas had lower coverage levels (Appendix A), but these were not densely populated areas, hence the zero-dose estimates were lower relative to some northern districts with similar coverage estimates. 

In 2018, the national estimates of DTP, MCV, and composite zero-dose children were 2,364,020, 3,121,156, and 1,703,296, respectively, while in 2021, these were 2,063,375, 2,784,980, and 1,457,068, respectively, indicating no increases in zero-dose prevalence due to the pandemic. The same pattern was generally observed at the district level (Figure 3), where we observed more decreases than increases in zero-dose prevalence (Figure 4). Additionally, we observed no (marked) increases in zero-dose prevalence in districts that had moderate to higher numbers of zero-dose children in 2018 (Figure 4d–f). However, there were a few districts, particularly those in the Lagos (Alimosho), Bauchi (Ningi), Kano (Ugongo, Dala, Tarauni, Kumbotso, Dawakin Tofa, Minjibir, Gwale, etc.), and Borno (Jere) states where considerable increases (>3000 unvaccinated children) were observed relative to the pre-pandemic period (Figure 4a–c). Also, we observed greater increases in zero-dose prevalence for MCV relative to DTP and the composite coverage indicator (Figure 3). 

The districts with the highest numbers of composite zero-dose children (>10,000) were mostly located in Zamfara state (Bungudu, Gusau, Kaura Namoda, Zurmi, Maradun, Maru and Bukkuyum) in 2018, whereas in 2021, these (>9000 zero-dose children) were located in the Bauchi (Ningi, Shira, Ganjuwa), Lagos (Alimosho), Kano (Ugongo), and Sokoto (Dange-Shuni) states, reflecting areas with a lack of access to or poor utilization of vaccination services in both time periods. In Appendix A, we display the zero-dose estimates at the state and regional levels to facilitate comparisons at these administrative levels. These estimates show that the greatest numbers of unvaccinated children were located in the northwestern region in both time periods and across all three indicators, driven by highly populated states such as the Kano and Katsina states. 

The zero-dose estimates are also displayed using interactive web-based maps (https://data.worldpop.org/repo/lf/visual/justice/district_number_zero_dose_DHS_MICS/ (accessed on 28 September 2023)) for better visualization.

### 3.4. Risk Factors Associated with Zero Dose at the National and Regional Levels before and during the Pandemic

The associations between the risk factors (adjusted odds ratios (aORs) and corresponding 95% credible intervals (CIs)) and the odds of vaccination or zero dose are plotted in Figure 5 and Appendix A at the national level for both time periods. When considering the composite coverage indicator (Figure 5), we observed strong similarities as well as subtle/minor differences in both time periods with respect to the factors associated with zero dose. Factors associated with the odds of zero dose in both time periods include: skilled birth attendance, birth quarter, mother’s receipt of tetanus toxoid vaccination status, mother’s education, ethnicity, household wealth, and access to bank account. The directions of the estimated relationships were generally the same between both time periods for these factors, except for the birth quarter, suggesting different seasonal patterns in vaccination in both periods. These similarities mostly reflect a lack of changes in the associations between the risk factors characterising maternal access to and utilization of health services, socioeconomic/demographic status, and the odds of zero dose in both time periods. Factors associated with zero dose in the pre-pandemic period in 2018 only include antenatal care attendance, access to media, use of the phone/internet, rural/urban, and travel time to the nearest health facility; while those associated with zero dose in the pandemic period only include marital status, livestock density, distance to coastline, and distance to conflicts. Considering that similar variables were also associated with vaccination in both time periods in most cases, these differences mostly reflect changes in the effect of communication, which was only associated with zero dose before the pandemic, and different characterizations of the effect of remoteness on vaccination in both time periods. We also note that the unexpected direction of the effect of the urban/rural variable before the pandemic is likely due to undetected collinearity or the effect of suppressing variables [7,48]. Detailed results of the estimated odds ratios are provided in the Appendix A (see Appendix A).

In Figure 5 and Figure 6, we provide summary plots of the (significant) risk factors that characterized the inequities in vaccination coverage at both the national and regional levels before and during the pandemic. At the national level, we found that the mother’s receipt of the tetanus toxoid vaccination, household wealth, access to a bank account, and the mother’s education were associated with all three zero-dose indicators in both the pre-pandemic and pandemic periods. Also, the mother’s ethnicity, religion, marital status, antenatal care attendance, and skilled birth attendance were associated with the receipt of DTP1 during both time periods, whereas the length of stay, mother’s age, and birth quarter were additionally associated with MCV1 in both periods. Additional factors associated with the composite coverage indicator in both periods were ethnicity, birth quarter, and skilled birth attendance. When examining the differences in the risk factors associated with zero dose/vaccination between both time periods, we observed that travel time and phone/internet use were associated with all three outcome indicators before the pandemic, while the distance to coastline was associated with all three outcomes during the pandemic. There were also other factors associated with vaccination in one time period only when examining individual outcome indicators—e.g., additional remoteness variables such as distance to the edge of cultivated areas and distance to conflicts were associated with DTP1 during the pandemic only. Interestingly, the mother’s age and length of stay (although with changing patterns for this risk factor) were only associated with MCV1 in both time periods and not associated with any other indicator in either or both time periods, highlighting the importance of both factors for MCV vaccination. Overall, these results agree with our initial conclusions via the composite coverage indicator.

At the regional level (Figure 6 and Appendix A), no risk factor was associated with all three indicators in both time periods in all the regions, evidencing greater variation in the associations between the risk factors and vaccination at the regional level, or the effect of the smaller sample sizes available at this level in the analysis (as highlighted previously, these analyses at the regional level were undertaken with a reduced set of risk factors due to sample size limitations). However, when examining individual indicators, we found that the mother’s education was associated with DTP zero dose in all four regions in both time periods. Also, the mother’s education and the mother’s age were associated with MCV zero dose in all four regions in both time periods. No risk factor was associated with composite zero dose in all four regions and both time periods, although there were strong effects of religion in the northern regions, the mother’s education in the northeast and southern regions, and the mother’s receipt of the tetanus toxoid vaccination in the north central and southern regions.

In the northwestern region, which had the highest estimates of the numbers of zero-dose children among the six regions, no risk factor was associated with all three indicators in both time periods, although the mother’s education was associated with DTP1 and MCV1 in both time periods. Additional factors associated with all three indicators in either of the two time periods were antenatal care attendance (pre-pandemic), phone/internet use (pre-pandemic), access to a bank account (pandemic), and proximity to borders (pandemic). These results additionally evidence changes in the effect of communication between the two periods and the importance of maternal education, as in the national-level results. In the northeast region, the mother’s education and religion were associated with all three indicators both before and during the pandemic. Additionally, antenatal care attendance was associated with all three indicators in the pre-pandemic period in this region. No risk factor was consistently associated with all three indicators in one time period only. We note that the estimated associations in both periods for this region are also in agreement with the results obtained at the national level. 

In the north central region, no risk factor was associated with all three indicators in both time periods. However, the mother’s education and religion were associated with DTP and MCV zero dose in both time periods while the mother’s receipt of the tetanus toxoid vaccination was associated with DTP1 and composite coverage in both time periods. Other risk factors associated with all three indicators either before or during the pandemic were antenatal care attendance (pre-pandemic), household wealth (pandemic), livestock density (pandemic), and proximity to borders (pandemic). These results obtained for this region are generally in agreement with the national-level results, but additionally demonstrate the effect of wealth and potential cross-border migration on zero dose. In the southern region, the mother’s education was associated with all three indicators in both time periods. Additionally, travel time was associated with DTP and composite coverage zero dose in both time periods. Skilled birth attendance and access to a bank account were also associated with all three indicators before the pandemic. No risk factor was associated with all three indicators in the pandemic period only. These results are also in agreement with the national results and additionally highlight the effect of maternal literacy, remoteness, maternal access to and utilization of health services, and socioeconomic disparities on the odds of zero dose. 

In general, these regional-level results, though limited by smaller sample sizes, generally corroborate the findings at the national level and have additionally highlighted the risk factors most important for each region through consistent associations with all three indicators either in one or both of the time periods studied. 

## 4. Discussion

By evaluating recent spatial and temporal trends in the distribution of zero-dose children in the context of the COVID-19 pandemic, our study further strengthens the scientific evidence base for improving childhood immunization in Nigeria. 

Our study provided estimates of numbers of unvaccinated children for DTP, MCV, and a composite coverage indicator at different spatial scales during the pre-pandemic and pandemic periods in Nigeria. Interestingly, our 2018 national-level DTP and MCV zero-dose estimates of 2.4 million and 3.1 million are in very good agreement with the (WHO and UNICEF estimates of national immunization coverage) WUENIC zero-dose estimates of 2.2 million and 3.1 million, respectively. Also, our 2021 national-level DTP and MCV zero-dose estimates of 2.1 million and 2.8 million in 2021 are very close to the corresponding WUENIC zero-dose estimates of 2.2 million and 2.9 million children, respectively (WUENIC zero-dose estimates were calculated using 2022 WUENIC coverage estimates and the UNPD estimates 2022 revision). Clearly, the pandemic did not result in any dramatic increases in zero-dose prevalence at the national level, but the persistence of large numbers of unvaccinated children in both time periods means that renewed efforts and novel strategies are needed to reach zero-dose and missed communities in the country. At the district level, no dramatic increases in zero-dose prevalence were found during the pandemic relative to the pre-pandemic era. However, there were some areas with elevated zero-dose estimates (>3000 children) during the pandemic, as highlighted previously. Some of these districts were located in the Kano and Lagos states where either relatively higher COVID-19 cases or deaths [49] were recorded during the study period, which could have also occurred as a result of the larger population sizes of both states [38]. The subnational variation in the effect of the pandemic on zero-dose prevalence in Nigeria has also been reported at the state level in a previous study [31], which focused on the Kano and Kaduna states. We note that the lack of substantial increases in zero-dose prevalence at the national level and in many subnational areas in our study, contrary to expectations, might have been due to modest interruptions or a quick recovery from the disruptions caused by the pandemic [50,51]. Additionally, our study revealed strong geographical disparities and a clear north–south divide in zero-dose prevalence in both time periods and across all three indicators, with districts with higher numbers of zero-dose children concentrated in the northern areas, as corroborated by previous studies [14,15,20,36,52]. However, there were also some districts in the south (e.g., in Lagos state) with higher numbers of zero-dose children. This recurring spatial pattern in the distribution of zero-dose children is a strong indication that targeted RI and campaign strategies, focusing on the most problematic areas, will be needed to achieve substantial reductions in the zero-dose prevalence within the country. Previous studies [20,21] have also revealed higher measles case counts in the north and high correlations between measles case counts and MCV zero-dose estimates, further strengthening the evidence for targeted interventions.

The underlying coverage levels also had similar patterns, revealing persistent areas of low coverage, mostly concentrated in the northeast and northwest regions across all three indicators and for both time periods. There were also persistent pockets of low coverage areas in the south, e.g., some areas in Cross River state and some areas near the coastline. However, we note that there were differences in the problematic areas when examining coverage and the zero-dose estimates at the district level. For example, there were some districts in the Lagos and Ogun states with moderate coverage levels, but which had higher zero-dose estimates. Also, some of the low coverage districts in Cross River state did not have higher zero-dose estimates, likely due to these areas having lower population densities. Hence, efforts aimed at reducing zero-dose prevalence should target areas where higher zero-dose estimates were estimated, whereas strategies to improve the equity in coverage should focus on the low coverage areas. When comparing maps of DTP1 and MCV1 coverage, we observed very similar patterns, with DTP1 coverage being higher in many places, due to the dropouts that often occur between both vaccine doses (and perhaps, the result of the suspension of MCV campaigns during the pandemic in 2021). This is a strong indication that the frequent campaigns conducted in Nigeria for MCV, though an effective temporary measure, have not been successful in boosting coverage beyond RI levels. The targeted strategies advocated earlier should, therefore, focus more on strengthening the country’s RI program, as we have also argued elsewhere [20]. Furthermore, when examining maps of the composite coverage indicator, the low coverage areas occurring mostly in the northeast and northwest and overlapping considerably with low coverage areas for MCV1 and DTP1, are strongly indicative of the non-availability of vaccination services and/or vaccine hesitancy. Different strategies would be required in these areas to unravel and address the barriers to vaccination. 

When examining the risk factors associated with zero dose, we found that while there were strong similarities between the pre-pandemic and pandemic periods, there were also some minor differences, which appeared more pronounced at the subnational/regional level. These similarities and differences are important for characterising the inequities that exist in the vaccination coverage in both time periods. At the national level, our study revealed consistent associations between each of the socioeconomic status (e.g., maternal literacy, household wealth and access to a bank account) and maternal access to and utilization of health services (e.g., skilled birth attendance) and the odds of zero dose in both time periods. We also found evidence of consistency in the effect of demographic factors (e.g., ethnicity, religion, and the mother’s age) and the seasonality of vaccination (e.g., birth quarter) on the odds of zero dose in both time periods. At the regional level (based on a reduced set of risk factors), we found additional evidence supporting the results obtained at the national level. Also, these regional-level analyses revealed the risk factors most relevant to reaching zero-dose and missed communities in each region. These were maternal access to and utilization of health services (all regions), communication (northwest), socioeconomic status (northwest, northcentral, and south), religion (northeast and, to a great extent, north central), cross-border migration (northwest and northcentral), and remoteness (south). Furthermore, at the national level, we did not find any remarkable differences in the associations between the risk factors and the odds of zero dose between both time periods. However, we found that there were changes in the variables characterizing the effect of remoteness on zero dose in both time periods. For example, travel time to the nearest health facility was associated with all three zero-dose indicators before the pandemic, while distance to coastline was associated with all three zero-dose indicators during the pandemic. Also, there was a pronounced positive effect of communication on the odds of vaccination before the pandemic, suggesting reduced communication regarding vaccination services during the pandemic. We did not explore the differences between both time periods at the regional level further due to the smaller sample sizes at this level. 

To facilitate the operationalization of these findings, our study produced interactive web-based maps (https://data.worldpop.org/repo/lf/visual/justice/all_indicators_coverage_final/; https://data.worldpop.org/repo/lf/visual/justice/district_number_zero_dose_DHS_MICS/ (accessed on 28 September 2023)) to further assist with the identification of towns, communities, and, potentially, settlements in the problematic areas. Additional analyses can also be undertaken through triangulation with other data sets, e.g., data on public health facilities offering vaccination services, to better understand the costs and/or efforts needed to reach zero-dose children within each district. Furthermore, the multi-temporal analyses presented here are highly relevant to planning effective outbreak response strategies or catch-up vaccination activities. Nigeria is currently experiencing a diphtheria outbreak which, according to reports [19], has been more pronounced in the Kano, Katsina, Yobe, Bauchi, Kaduna, Borno and Jigawa states as of the beginning of October 2023. Interestingly, these states were among the states where we had estimated the highest prevalence of DTP zero-dose children in both 2018 (mostly between 120,000 and 215,000 DTP zero-dose children per state—see Appendix A) and 2021 (mostly between 80,000 and 240,000 DTP zero-dose children per state), further corroborating the findings from our study. Also, the occurrence of a considerable proportion (one third) of the confirmed cases of the disease in children aged between 5 and 9 years (as of October 2023), which includes the birth cohort for which we produced zero-dose estimates in 2018 in our study, further evidences the programmatic and operational relevance of our analyses. Specifically, our maps of DTP zero-dose children for both years can be used to determine areas where interventions are needed to fill immunity gaps in both older and younger birth cohorts throughout the country. We also note that our district/LGA-level zero-dose estimates can be further disaggregated to the ward level and health facility catchment areas to enhance field operations if needs be.

Through its Zero-dose Reduction Operational Plan (Z-DROP) programme, Nigeria is continuing to intensify efforts to reach its zero-dose and missed communities. Fundamentally, the Z-DROP programme is one of the strategies for achieving the country’s vision of integrated primary health care service delivery [53]. Through a rigorous prioritization exercise led by the National Primary Health Care Development Agency, Gavi, the Vaccine Alliance, and the University of Southampton in August 2022, about 100 LGAs were identified as priority areas where (RI) interventions were urgently needed to reach zero-dose and under-immunized children. About 60 of these LGAs, spread across eight states, are being targeted in the current phase of the Z-DROP programme. The programme employs a bottom-up approach to design and implement interventions in these areas through engagement with local health workers. These interventions include initial catch-up immunization activities planned as part of the 2023 measles campaigns, aiming to administer recommended routine vaccines to identified zero-dose children, and then follow up RI activities to sustain the gains made and to ensure the completion of the immunization schedule. The process of identifying zero-dose children in these LGAs additionally involves the triangulation of coverage survey/zero-dose, surveillance, and outreach services data at the ward and health facility levels to identify, geolocate, and classify (unreached, far-to-reach, hard-to-reach, and never reached) high-priority settlements. These additional analyses also include estimating the target populations and the cost of implementing the required interventions in the identified high-priority settlements to guide resource allocation. The programme also provides a mechanism to document all operational activities for the effective supervision and timely tracking of progress.

Our study is subject to some limitations. Our vaccination coverage estimates were produced using information obtained from both home-based records and maternal/caregiver recall, with the latter being subject to recall bias. The sampling frames used in both the 2018 DHS and 2021 MICS-NICS may have missed important vulnerable populations such as those living in conflict areas in Borno state, as highlighted previously. This may have led to an underestimation of the zero-dose prevalence in some areas. Our analyses included comparisons of the vaccination coverage and zero-dose estimates between the 2018 NDHS and 2021 MICS-NICS to assess the impact of the COVID-19 pandemic on immunization services in Nigeria. Since these surveys were implemented independently and not as repeated or rolling surveys, differences in the survey instruments (e.g., questionnaires), sampling designs, and implementation, could have affected the differences seen in the comparisons. Our analyses utilized displaced cluster-level geographical coordinates to predict coverage levels at a 1 × 1 km resolution. While this may not matter for coverage and zero-dose estimation at the district level using the 2018 DHS data, since the DHS program often retains the displaced clusters within their original districts [54], the displacement may have affected the district-level estimates produced using the 2021 MICS-NICS, as the initial displacement conducted by the MICS team which was used in our work only preserved the state boundaries. Since completing our analyses, the displacement of the geographical coordinates from the 2021 MICS-NICS has been updated to preserve the district-level boundaries. We carried out some sensitivity analyses (results not presented here) using the updated coordinates, which revealed very minor differences from the results (coverage maps) presented in this work. Furthermore, we did not quantify the uncertainties associated with the zero-dose estimates presented in our work. These uncertainties can arise from both the vaccination coverage and population estimates. In particular, the population estimates used in our work were based on projections from the 2006 Nigeria population and housing census and broad area-level age breakdowns. Also, no uncertainty estimates were available for these estimates since they were produced using the “top-down approach” [38]. However, alternative approaches [55] can be used to produce more accurate population estimates and associated uncertainties when recent input data are available. Also, when uncertainty estimates are available for the population estimates [55,56], these can be combined with the uncertainties from vaccination coverage in a statistical framework to produce uncertainties for the zero-dose estimates. Our analysis of the risk factors associated with zero dose included mainly demand-side factors due to data limitations. The inclusion of supply side factors in future work will likely yield more programmable insights and will further explain any residual variation in the multilevel models for the coverage indicators. Lastly, our exploration of the differences in the risk factors associated with zero dose at the regional level in the pre-pandemic and pandemic periods was limited by the smaller sample sizes. This challenge can be overcome in future work through a pooled data analysis. 

As immunization programmes around the world continue to recover from the disruptions to immunization services caused by the COVID-19 pandemic and get back on track to achieving the goals and targets set out in the Immunization Agenda 2030, our study has provided programmatically important insights that can aid policy makers to plan and implement effective strategies to reach zero-dose and missed communities in Nigeria. 

## Figures and Tables

**Figure 1 vaccines-11-01830-f001:**
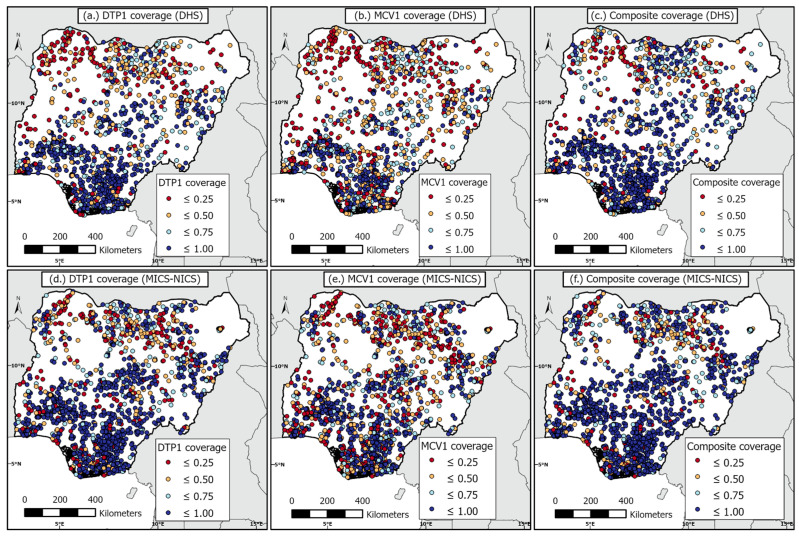
Survey cluster locations and observed vaccination coverage for children aged 12–23 months for both the 2018 DHS (**top** panels) and 2021 MICS-NICS (**bottom** panels).

**Figure 2 vaccines-11-01830-f002:**
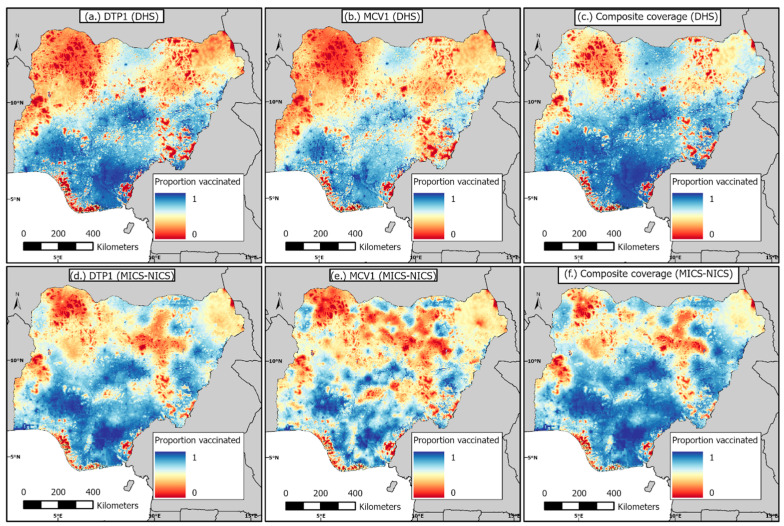
1 km × 1 km modelled estimates of vaccination coverage for DTP1, MCV1, and composite coverage indicator before the pandemic, produced using the 2018 DHS (**top** panels), and during the pandemic, produced using the 2021 MICS-NICS (**bottom** panels). The associated uncertainty maps are shown in Appendix A.

**Figure 3 vaccines-11-01830-f003:**
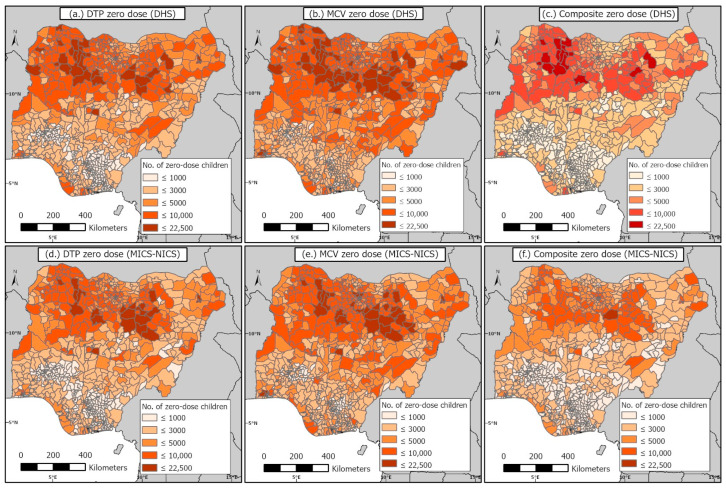
District-level estimates of numbers of DTP, MCV, and composite (i.e., BCG, OPV, DTP, and MCV) zero-dose children during the pre-pandemic period in 2018 (**top** panels) and the pandemic period in 2021 (**bottom** panels). Corresponding coverage estimates are shown in Appendix A.

**Figure 4 vaccines-11-01830-f004:**
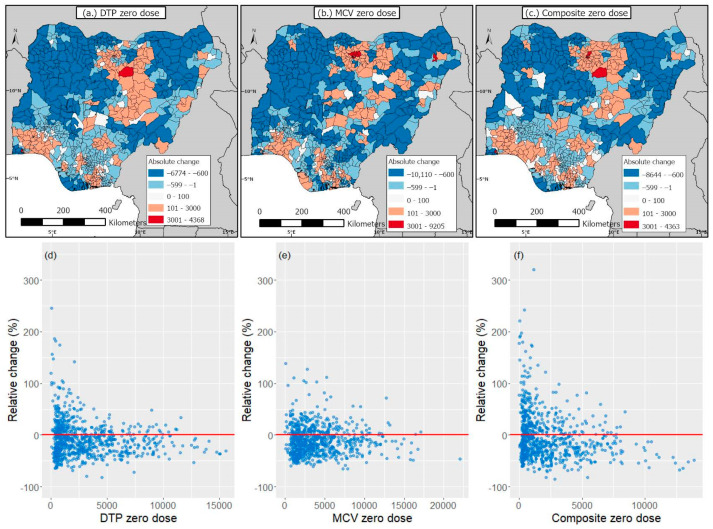
(**a**–**c**) Absolute changes in estimates of numbers of zero-dose children between 2018 and 2021 (i.e., 2021 estimates minus 2018 estimates) and (**d**–**f**) 2018 zero-dose estimates versus the relative changes in zero-dose estimates between both time periods.

**Figure 5 vaccines-11-01830-f005:**
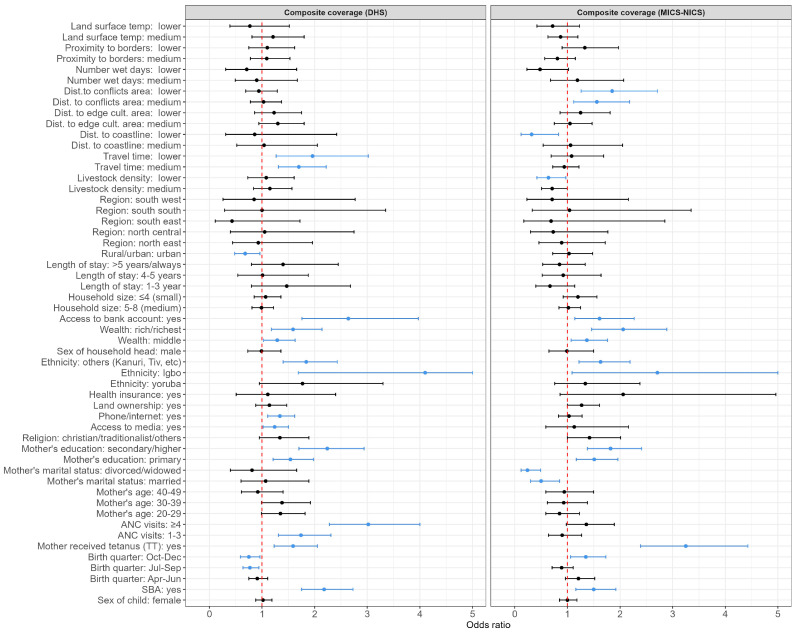
Adjusted odds ratios (aORs) and corresponding 95% credible intervals (95% CI) showing associations between the risk factors and the composite coverage indicator in the pre-pandemic period in 2018 (DHS) and the pandemic period in 2021 (MICS-NICS) at the national level. The vertical dotted red lines mark the odds ratio of 1. Light blue dots and lines show the aORs and 95% CIs of variables that had significant associations with zero dose. The black dots and lines show the aORs and 95% CIs of variables that had no significant associations with zero dose. Some upper CIs have been truncated at a value of 5. The definitions of the risk factors and their reference categories are provided in Appendix A.

**Figure 6 vaccines-11-01830-f006:**
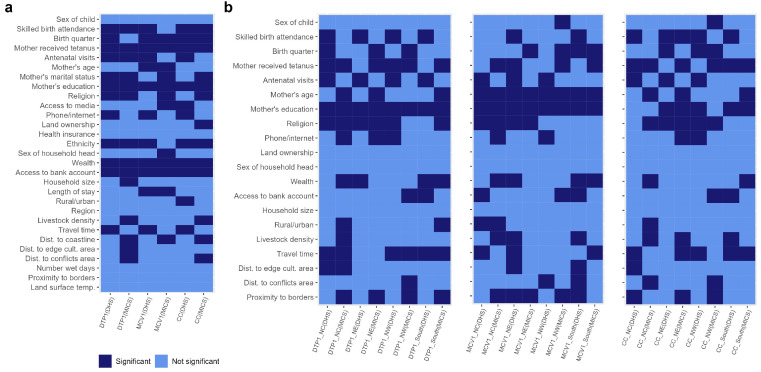
Summary plots showing the risk factors associated with zero dose before the pandemic in 2018 (DHS) and during the pandemic in 2021 (MICS-NICS) at the (**a**) national and (**b**) regional levels, identified using three vaccination coverage indicators, namely DTP1, MCV1, and composite coverage (CC).

## Data Availability

The data used in this study are available from the DHS (https://dhsprogram.com/data/available-datasets.cfm (accessed on 28 September 2023)) and MICS (https://mics.unicef.org/surveys (accessed on 28 September 2023)) programmes upon request. Other data (i.e., geospatial covariates) are publicly available via the sources referenced in the manuscript. The authors are not allowed to redistribute these datasets.

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
