# Peer review of "Geospatial Analyses of Recent Household Surveys to Assess Changes in the Distribution of Zero-Dose Children and Their Associated Factors before and during the COVID-19 Pandemic in Nigeria"

_vaccines, 2023, doi:10.3390/vaccines11121830_

Round 1

Reviewer 1 Report

Comments and Suggestions for Authors

Very well-done study with important topic. 

I only have one minor comment: how credible is the population estimate in Nigeria? I believe that the census has not been done for a while, and I am curious as to how the estimate can be trusted. If you could include some more descriptions / limitations, that would be great. 

Author Response

Reviewer #1:

Very well-done study with important topic. 

Response: Thank you for the positive feedback, appreciated.

I only have one minor comment: how credible is the population estimate in Nigeria? I believe that the census has not been done for a while, and I am curious as to how the estimate can be trusted. If you could include some more descriptions / limitations, that would be great. 

Response: Thank you. The last census in Nigeria took place in 2006. The population estimates used in this study, which were obtained from WorldPop (https://www.worldpop.org/), were produced using administrative-level projections from this census. The estimation methodology (a top-down approach -https://www.worldpop.org/methods/top_down_constrained_vs_unconstrained/) combined these projections with relevant and more up-to-date geospatial covariate information, to produce more accurate estimates of population distribution at the grid level. These estimates were further broken down by age and sex and adjusted to match the United Nations Population Division (UNPD) estimates at the national level.  We do note that the estimation process, as well as the estimates being based on population projections, means lots of uncertainties which we have now further clarified in the Discussion (page 17 lines 707-714, page 17). A different method known as the “bottom-up approach” can be used to produce more accurate population estimates when recent input microcensus data are available.

Reviewer 2 Report

Comments and Suggestions for Authors

This is a well-written article, keeping all the etiquette of research in spatial epidemiology.  This is a real value addition to two existing nationwide datasets in Nigeria, which will have policy implications to safeguard the lives of children in this resource-limited setting. However, in order to improve the readability and to reach a wider readership, I have the following comments.

1. Authors should give the full form of the abbreviations used at the first mention of those in the article.

2. The definition of the outcome measure "zero-dose children" should be given in the initial part itself.

3. The authors are basing a lot of interpretation on the figures and tables in the supplementary file, and that is missing in the review set given.  So the aptness of these illustrations could not be verified.

4. Both the interactive links (link1 and link2) are not working.

Comments on the Quality of English Language

In general, it is written in good English. My only request is to give the full forms of the abbreviations at the first mention of the terms itself.   This paper might attract people from many other disciplines, so the commonly used terms in medical literature also need expansion.

The following sentence can be verified, “Due to data availability, our study considered only demand-side factors or reasons for non-vaccination.”  I thought it might be meant for,  "Due to data constraints, ..."

Author Response

Reviewer #2:

This is a well-written article, keeping all the etiquette of research in spatial epidemiology.  This is a real value addition to two existing nationwide datasets in Nigeria, which will have policy implications to safeguard the lives of children in this resource-limited setting. However, in order to improve the readability and to reach a wider readership, I have the following comments.

Response: Thank you for the positive feedback on our paper.

  1. Authors should give the full form of the abbreviations used at the first mention of those in the article.

Response: Thank you. We have addressed this throughout the manuscript. For example, see lines 34, 40-42 in the abstract and line 119, page 3.

  1. The definition of the outcome measure "zero-dose children" should be given in the initial part itself.

Response: Thank you. We have provided relevant definitions of "zero-dose children" in the first (lines 71-72) and last paragraphs (line 143-148) of the Introduction.

  1. The authors are basing a lot of interpretation on the figures and tables in the supplementary file, and that is missing in the review set given. So, the aptness of these illustrations could not be verified.

Response: Thank you. We submitted the supplementary file including all the figures. We are sorry that this was not provided to you for review.

  1. Both the interactive links (link1 and link2) are not working.

Response: Thank you. We have cross-checked that the links are working correctly. The interested reader only needs to click the link in the PDF or hold down the control key on the keyboard and click the link in word document to access the interactive maps.

Comments on the Quality of English Language

In general, it is written in good English. My only request is to give the full forms of the abbreviations at the first mention of the terms itself.   This paper might attract people from many other disciplines, so the commonly used terms in medical literature also need expansion.

Response: Thank you. We have addressed this throughout the manuscript. For example, see lines 34, 40-42, page 1, line 67, page 2 and line 119, page 3.

The following sentence can be verified, “Due to data availability, our study considered only demand-side factors or reasons for non-vaccination.”  I thought it might be meant for, "Due to data constraints, ..."

Response: Thank you. We have revised the manuscript to reflect this. See line 148, page 3.

Reviewer 3 Report

Comments and Suggestions for Authors

See attached word document.

Author Response

Reviewer #3:

In this paper, the authors leveraged the data from household surveys conducted before and during the COVID-19 pandemic in Nigeria to investigate the potential effects of the pandemic on vaccination coverage among children, as well as associated socio-economic and demographic factors. They applied a robust Bayesian geostatistical framework to estimate vaccination coverage during two periods at different scales and multilevel models to identify risk factors of zero-dose vaccination before and during the pandemic. The geospatial analysis demonstrated clear geographical disparities in vaccination prevalence across the country. Several risk factors were identified, which would help design tailored vaccination strategies for specific areas in the future. This study was well designed, and the manuscript was interesting to read. I would suggest accepting it for publication after the following comments are addressed.

Response: Thank you. We appreciate your support for our paper and the helpful feedback.

General comments

  1. The Introduction section implies that your hypothesis was the coverage/prevalence of zero-dose vaccination increased during the COVID-19 pandemic, given the pandemic could disrupt the vaccination services in Nigeria. Your results suggested there was no increase in zero-dose prevalence, and you even observed more decreases in some districts. The prevalence maps in Figure 2 also show a similar decreasing trend, especially in the northwestern part where the prevalence has been persistently high. I was expecting to find some potential explanations on how the pandemic contributed to decreasing the zero-dose prevalence in some areas, based on the change/difference of risk factors before and during the pandemic that you identified from your models. However, I saw few in the Discussion. I think the readers will also be interested to know this. Some social patterns could have been altered because of the pandemic (e.g., travel time to health facilities), and, consequently, this change might play a positive role in increasing the vaccination coverage in some underserved communities in certain ways. Understanding this will also contribute to developing effective vaccination strategies in the future.

Response: Thank you for these helpful comments. As you rightly noted, our results did not show any remarkable increases in zero-dose prevalence due to the pandemic, contrary to expectations. We also found strong similarities in the risk factors associated with zero dose before and during the pandemic. Following previous studies referenced in the manuscript, we have mentioned in the Discussion (lines 577-581, page 15) that these findings might have been due to modest disruptions caused by the pandemic or a quick recovery from the disruptions (these studies established that there were varying amounts of disruptions using routine data and also found a quick rebound from these). We are unable to make any speculations outside of these as neither our data/analyses nor existing studies (except those that we may not be aware of) support this. Additionally, we have already mentioned in the limitations that the two surveys analyzed in our work were designed independently and not as rolling surveys, which may have affected the differences in our comparisons.

Specific comments

Introduction

  1. Line 68: are these 18.4 million children from LMICs? Please specify it.

Response: Thank you. This is at the global level. We have revised the manuscript to address this. See line 69, page 2. 

  1. Line 115: please spell out “RI” as it first appears in the text

Response: Thank you. The ‘RI’ stands for ‘routine immunization’. We have revised the manuscript to reflect this. See line 119, page 3.

Methods

  1. Line 283: the letter l was missing right after stratum

Response: Thank you for pointing this out. We have revised the manuscript to reflect this. See line 288, page 7.

  1. Line 315: I would suggest moving the ‘Geostatistical model’ section before ‘Multilevel model’ as you first presented the results from geostatistical models in the following section.

Response: Thank you for the suggestion. In the methods section, it is best to present the simpler model (multilevel) before moving into a more complex (geospatial) model. This makes for a natural progression in the description of the models used in the work. However, we decided to present the results of the geospatial analysis first before the multilevel results because that is the main/primary focus of our study, and the rationale was to use the results of the multi-level analyses to further explore/describe the patterns seen in the geospatial analyses results. We therefore wish to let the multilevel model section come before the geospatial methods section as presented in the original manuscript, and the results for the geospatial model to come first before the multilevel model results.  

Results

  1. Line 350-352: please remove the instructions.

Response: Thank you for pointing out this. We noted that this was removed by the Editors before our revision.

  1. Line 359: … appeared to be higher …” you can consider doing a T-test to compare the two years.

Response: Thank you for the suggestion. This statement was based on comparisons of national level coverage estimates between the two surveys analysed in our work, meaning that we only had one value/observation for each survey at each of the survey locations. In case the reviewer was thinking of us comparing the national means for DHS and MICS-NICS surveys, this is not appropriate because it will not reveal any spatial differences in the outcomes. Hence, we are unable to do a t-test in this case.

  1. Line 373: “… significant overlaps …” how did you identify the overlaps and measure the significance of that?

Response: Thank you. The identification of the overlaps was by visual inspection of the coverage maps as can be clearly observed in Figure 2, and by significant, we were not referring to statistical significance. For clarity, we changed the word ‘significant’ to ‘substantial’ and revised the manuscript accordingly. See line 374, page 8.  

  1. Line 385: “Table 0, …” the beginning of this sentence is not complete.

Response: Thank you. We believe that this error was introduced by the journal during journal word formatting and conversion of our word document file to PDF file for the review process because the word version we submitted did not have this omission. In our submitted word version, we have “The corresponding uncertainty maps showed standard deviations less than 0.33 for the predicted vaccination coverage estimates for 2018 DHS and less than 0.36 for the 2021 MICS-NICS, suggesting low uncertainties around the predicted coverage estimates in both years (supplementary Figure S3)” which is different from what was reported in the PDF reviewed. We revised the manuscript to address this. See line 387, page 9.  

  1. Line 430: Figure 3 - why did you choose to display the estimates of numbers but not coverage in Figure 3? If the estimates of coverage were shown here, the readers can directly compare them with Figure 2.

Response: Thank you for this feedback. In this figure, the main goal was to present the absolute numbers of zero-dose children based on each of the 3 indicators of vaccination coverage at the district level for both survey years. The corresponding district level vaccination coverage estimates are presented in Supplementary Figure S5 as we have already mentioned in the text (line 403, page 9) and the title of the figure. These district level coverage estimates have broadly similar patterns as the grid level estimates presented in Figure 2.